

# Structural monitoring for lifetime extension of offshore wind monopiles: Can strain measurements at one level tell us everything?

Lisa Ziegler[1,2], Ursula Smolka[1], Nicolai Cosack[1], Michael Muskulus[2]

[1]Ramboll Wind, 20097 Hamburg, Germany

[2]Department of Civil and Environmental Engineering, Norwegian University of Science and Technology NTNU, 7491 Trondheim, Norway

*Correspondence to*: Lisa Ziegler (lisa.ziegler@ramboll.com)

**Abstract.** Operators need accurate knowledge on structural reserves to decide about lifetime extension of offshore wind turbines. Load monitoring enables us to directly compare design loads with real loading histories of the support structure in

order to calculate its remaining useful lifetime. Monitoring of every hot spot is technically and financially not feasible. This paper presents a novel idea for load monitoring of monopiles. It requires strain measurements at only one level convenient for sensor installation, such as tower bottom. Measurements are converted into damage equivalent loads for 10-minute time intervals. Damage equivalent loads are extrapolated to other locations of the structure with a simulation model and statistical algorithm. For this, structural loads at all locations of the monopile are calculated with aero-hydro-elastic software and

updated finite element models. Damage equivalent loads at unmeasured locations are predicted from the simulation results with a k-nearest neighbor regression algorithm. The extrapolation was tested with numerical simulations of an 8 MW offshore wind turbine. Results show that damage can be predicted with an error of 1-3 % if this is done conditional on mean wind speed, which is very promising. The load monitoring concept is simple, cheap and easy to implement. This makes it ideal for better decisions on lifetime extension of monopiles.

**1 Introduction**

Load monitoring of foundations for offshore wind turbines enables to reconstruct load histories that these structures experienced. The load history can be compared against design loads to calculate remaining useful lifetimes, which is essential for decisions on lifetime extension. Direct monitoring of every hot spot at the structure is impossible due to cost and access restrictions. Structural responses must be extrapolated from a limited set of sensors.

81% of offshore wind turbines foundations were monopiles in 2016 (Ho and Mbistrova, 2017). Existing monitoring strategies for monopiles are based on physical models or artificial intelligence. Model-based time-domain algorithms require accelerometers and (partly) strain gauges at the structure. They try to reproduce the time history of dynamic response parameters, such as acceleration or strain, of the whole structure. This has been investigated for monopiles using Kalman filters (Maes et al., 2016; Fallais et al., 2016), joint input-state estimation (Maes et al., 2016), and modal expansion

algorithms (Maes et al., 2016; Iliopoulos et al, 2016). In many cases, the remaining useful lifetime can be assessed using



accumulated cycles or equivalent loads. Detailed load time series are not required. This is exploited by artificial intelligence algorithms (e.g. neural networks) (Smolka and Cheng, 2013; Cosack, 2011). After being trained using measurement data from all hot spots, the algorithms deduce statistics of dynamic response parameters, such as equivalent loads, from standard signals (Smolka et al, 2014).

For the decision on lifetime extension, the onshore wind industry uses numerical fatigue reassessment to recalculate structural loading with updated design models and assumptions (mainly environmental and operational conditions) (Ziegler et al., submitted; Ziegler and Muskulus, 2016). Drawbacks are that long-term measurements of some environmental conditions, such as turbulence intensity, are often not available or expensive to obtain. Load monitoring will be useful for lifetime extension, however, operators are still reluctant due to associated costs.

The authors developed a novel load monitoring concept that requires only minimal sensor placement. Load measurements at tower bottom are transformed into damage equivalent loads and extrapolated to other hot spots. This novel idea is presented in Section 2. Performance of the algorithm is discussed in Section 3 and concluded in Section 4.

## 2 Methodology

The methodology presented here requires to measure loads at only one location of the structure where installation and
maintenance is convenient, such as near tower bottom. This information is used to predict damage equivalent loads (DELs) at all relevant hot spots of the monopile. Figure 1 (right) illustrates the setup.

*DEL* is defined as the single-amplitude load (or stress) range that causes the same amount of damage over a reference number of cycles $N_k$ as the variable-amplitude load (or stress) time series $S_i$ with corresponding number of cycles $N_i$ (cf. Equation 1). Here $n$ is the number of stress ranges, and $m$ is the inverse slope of the considered SN-curve (DNVGL, 2016).
Further information on DELs can be found in (Cosack, 2011).

$$DEL = \left( \sum_{i=1}^{n} \frac{N_i}{N_k} S_i{}^m \right)^{\frac{1}{m}} \tag{1}$$

The methodology consists of the following steps:

1. The finite element model of the monopile from the design phase is updated (e.g. with short-term on-site measurements) to ensure consistent dynamic behavior.
2. Aero-hydro-elastic simulations are performed with the updated model and an extended design basis. Simulation outputs are 10-min load (or stress) time series at the measurement location and at all locations of interest. Rainflow counting is performed on these time series and DELs are calculated for all hotspots of interest.
3. The transfer function between hotspots and measurement location are calibrated using simulated DELs. Details on the model are given in Section 2.1.
4. The load measurements are converted into 10-minute DELs. The extrapolation model is used to predict the DELs at other locations of the structure.





5. Calculation of accumulated fatigue damage $D$ at all desired locations and remaining useful lifetime with Equation 2 and 3. Here $a$ is the value of the SN-curve at $N_k$ cycles and $t_{op}$ is the number of year the wind turbine has operated already.

$$D = \frac{DEL^m N_k}{a} \tag{2}$$

$$RUL = \frac{t_{op}}{D} - t_{op} \tag{3}$$

## 2.1 Extrapolation model

The relationship between DELs at tower bottom (T-DELs) and other locations of the monopile is assumed to be well defined. The extrapolation to DELs at mudline (M-DELs) is investigated as an example in the following. In Figure 1 (left, top) T-DELs from aero-hydro-elastic simulations are plotted in ascending order for 1700 load cases. The corresponding

10 M-DELs are shown as black dots. Each load case has different inputs in terms of mean wind speed, turbulence intensity, sea state, wind and wave directionality, and operational condition.

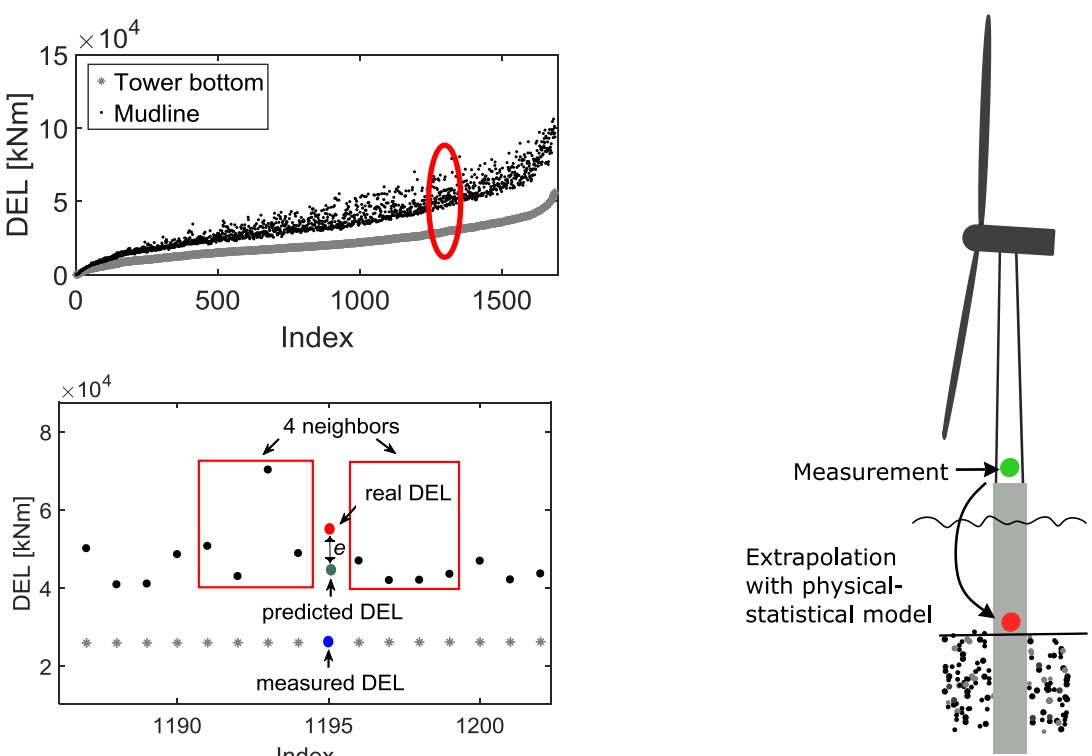

**Figure 1. Left top:** DELs at tower bottom (T-DELs) sorted ascending for 1700 load cases. Corresponding DELs at mudline (M-DELs) are plotted as function of T-DELs. **Left bottom:** Zoom into picture above. The M-DEL (green dot) corresponding to a measured T-DEL (blue dot) is estimated as mean value of four nearest neighbor values (from simulations) on each side. It deviates from the real M-DEL with an estimation error $e$. **Right:** Schematic setup of load monitoring concept. Measured T-DELs are extrapolated to other locations.





Figure 1 (top, left) shows a well-defined lower bound for M-DELs. Scatter of M-DELs above this curve is limited. The highest M-DEL is only a factor of 2.3 higher than the lowest M-DEL for similar T-DELs. The observed small scatter of M-DELs enables to use a simple statistical model for T-DEL extrapolation (cf. Figure 1):

1. Sort a measured T-DEL into the array of simulated T-DELs.

2. Select a number of simulated T-DELs close to the measured T-DEL value (nearest neighbors).

3. Predict desired M-DEL as mean or a weighted mean of the simulated M-DEL values corresponding to the nearest neighbor simulated T-DELs. Weighting can be done with occurrence probability of simulation load cases, when statistics from the site are available.

10  This methodology is an application of the k-nearest neighbors regression algorithm from machine learning (Øye, 1999).

## 2.2 Accuracy and choice of neighbors

The accuracy of the extrapolation model is validated against simulation data here, as measurements were not available at this project stage. We use leave-one-out cross validation to assess the performance: One simulation result is dismissed (considered as "measured T-DEL"), the corresponding M-DEL is extrapolated with the remaining simulations, and then
15  compared with the value known from simulations.

Figure 2 shows extrapolated M-DELs in red plotted over simulated M-DELs for one neighbor (left) and for ten neighbors on each side (right). An increase in the number of neighbors causes smoothing of extrapolation results.

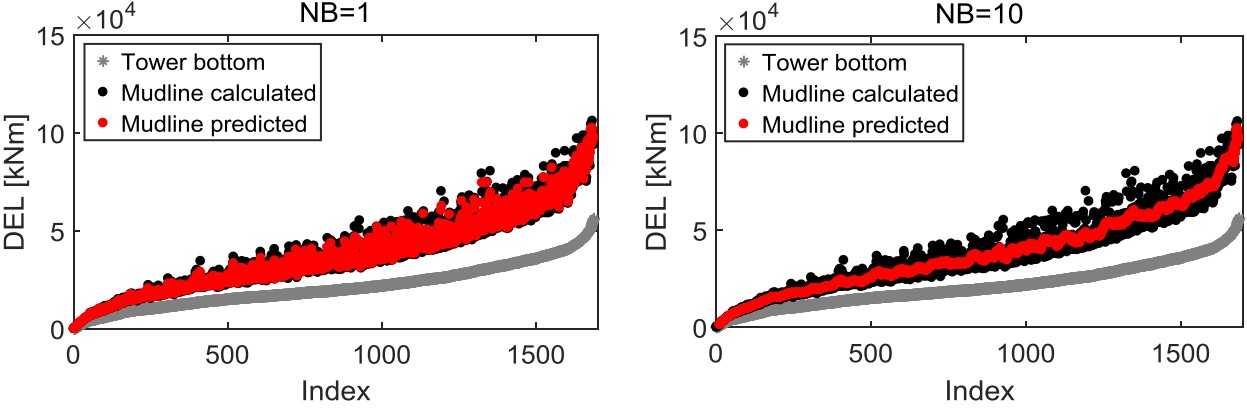

**Figure 2.** Simulated (black dots) and extrapolated M-DELs (red dots) considering one neighbor (left) and ten neighbors (right) under leave-one-out cross-validation.

The number of neighbors should be chosen so that damage at the end of service life is predicted best. Figure 3 shows a
20  measure of the estimation error of accumulated damage as a function of number of neighbors (solid line). Damage ratio are calculated as fraction "extrapolated damage / simulated damage". The dashed line indicates extrapolation uncertainty. It represents the deviation of damage from the desired result when the standard error of the mean of the neighbors is added to





the mean value during M-DEL extrapolation (cf. Eq. 4). $\sigma$ is the standard deviation of the sample of considered neighbors, $n_b$ is number of neighbors.

$$\Delta MDEL = \frac{\sigma}{\sqrt{n_b}} \qquad (4)$$

The damage ratio converges to 1.08 after four neighbors in this example (design basis, no binning, mean of neighbors – cf.
Section 3). The deviation of lifetime damage decreases for increasing number of neighbors, but beyond four neighbors the extrapolation accuracy seems insensitive to the number of neighbors used. The true value of 1.0 is inside the interval of two standard deviations (not plotted). The variance of the extrapolation error of individual DEL values increases with the number of neighbors (cf. Figure 3 right).

Available data from the turbine control and performance monitoring system (SCADA) can provide additional information for improving the extrapolation. Potentially relevant parameters are average wind speed, wind direction, and operational condition (power production or idling). To utilize this information, the simulated DELs are binned according to these parameters. Only DELs that have similar conditions (i.e., are from the same bin) are considered as neighbors in the extrapolation.

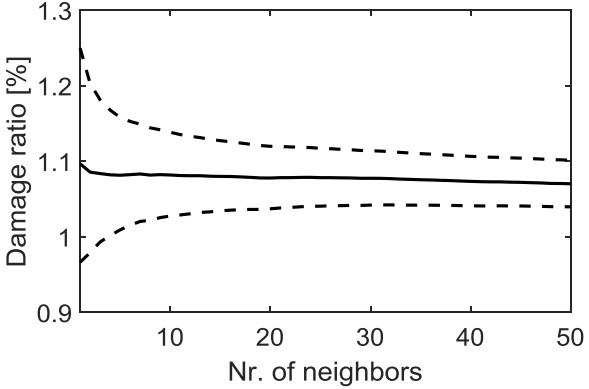
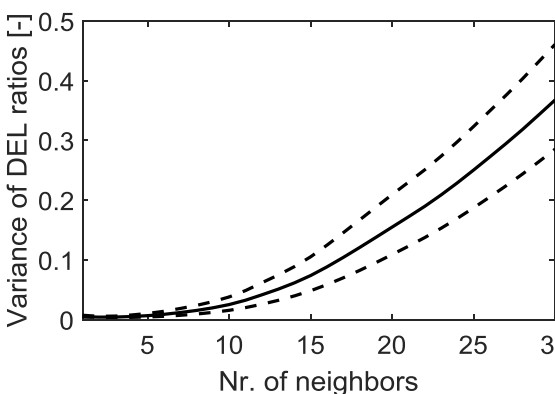

**Figure 3. Left:** Estimation error of lifetime damage as function of number of neighbors considered in the extrapolation algorithm. **Right:** Variance of individual DEL ratios as function of number of neighbors. DEL ratios are calculated as "extrapolated M-DEL / simulated M-DEL".

## 3 Results and discussion

### 3.1 Case study

Results are presented for a case study of a monopile in 30-40m water depth supporting an 8 MW wind turbine. Turbine and monopile were modelled in detail following industry state-of-art. Sequentially integrated load simulations were performed with the software LACflex and ROSAP (in-house tools of Ramboll).



LACflex is an aero-elastic software for time-domain analysis of wind turbines based on the solver FLEX 5 (Passon and Branner, 2014). ROSAP is a structural analysis program which Ramboll uses for design of offshore wind foundations. The detailed model of the monopile is reduced to a Craig-Bampton superelement including corresponding wave loads for accurate integration into LACflex. Response time series at tower bottom are imported into ROSAP to model hydrodynamic

loading and structural response of the detailed finite element model of monopile and transition piece (Passon and Branner, 2014; Passon, 2015). Design simulations of 10 min duration were performed for the fatigue load cases power production and idling according to IEC 61400-3 (IEC, 2009).

The extrapolation model is tested in four cases:

   1.   design basis,

2.   extended simulations,

   3.   design basis with wind speed binning, and

   4.   extended simulations with wind speed binning.

The design basis includes 1700 load cases of normal operation and idling with wind speeds from 2 m/s to 32 m/s and corresponding sea states and turbulence intensity. Wind-wave directionality is considered in 30° bins including

misalignment. The extended simulations include the design basis and additional 1700 load cases with reduced turbulence intensity. M-DELs for design basis (black) and lower turbulence intensity (red) are plotted in Figure 4 (left). The extended simulations follow the same pattern as the original set.

In Figure 4 (right) M-DELs are colored according to their mean wind speed. The lower bound observed for M DELs is driven by wind speeds below rated power (12 m/s). The high end of the scatter occurs predominantly for load cases above

cut-out wind speed (24 m/s).

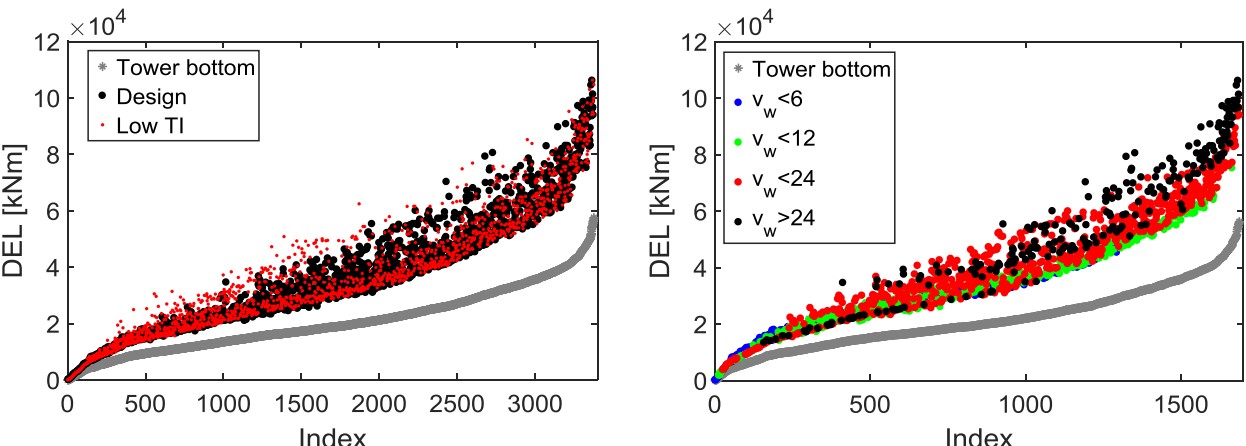

**Figure 4.** M-DELs as function of ascending T-DELs. **Left:** M-DELs for design basis (black) and lower turbulence intensity (red). **Right:** Design basis M-DELs are colored according to the input mean wind speed.



## 3.2 Extrapolation results

Table 1 presents the results of the extrapolation model in the four test cases obtained with leave-one-out cross-validation. Ratios are calculated as fraction "extrapolated value / simulated value" to indicate extrapolation errors. Figure 5 shows a histogram of 1700 DEL ratios for the case "design", where M-DELs are extrapolated as mean of 1 and 15 neighbors. It

resembles the shape of a normal distribution for one neighbor, while the skewness increases for more neighbors, resulting in some overprediction of the average.

Variance of the ratios is below 10% in all cases. The target values, lifetime M-DELs and damage, are extrapolated with errors smaller than 3% and 9%, respectively. The extrapolation error is larger for damage than for lifetime DELs due to exponentiation with the material parameter m. Extrapolating $DEL^m$ instead of DEL did not yield better results in this study

(not shown). Weighting with the probability of occurrence and wind speed binning improves damage extrapolation, leading to errors smaller than 1% and 3%, respectively.

**Table 1.** Results for M-DELs with 15 neighbors in four test cases. M: mean, W: weighted mean, $DEL_L$: lifetime M-DEL.

| Case | Method | Variance Ratio DEL [-] | Ratio $DEL_L$ [-] | Ratio Damage [-] |
|---|---|---|---|---|
| Design | M | 0.07 | 1.02 | 1.08 |
| | W | 0.06 | 1.00 | 0.99 |
| Design with $v_w$ bin | M | 0.01 | 0.99 | 0.97 |
| | W | 0.01 | 1.00 | 0.99 |
| Extended | M | 0.02 | 1.03 | 1.09 |
| | W | 0.02 | 0.99 | 0.97 |
| Extended with $v_w$ bin | M | 0.05 | 0.98 | 0.94 |
| | W | 0.08 | 0.99 | 0.97 |

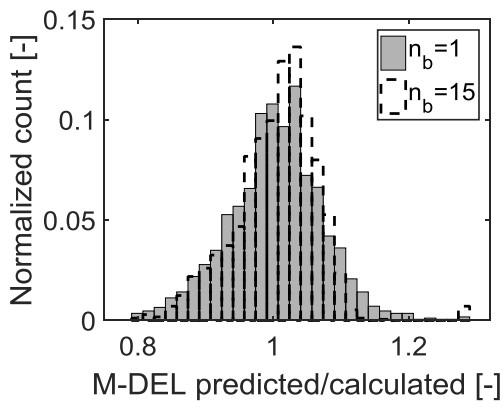

**Figure 5.** Histogram of DEL ratios for the case "design" with 1 and 15 neighbors.

## 3.3 Discussion

The novel idea for load monitoring is simple and easy to implement. No additional sensors, apart from strain measurements

at one level, are needed. Data transfer and storage is efficient as the algorithm works with 10-minute values after conversion to DELs.

Little costs of this solution make it feasible for application at every turbine in a wind farm. This removes the uncertainty of interpolation between turbines. DELs are calculated for SN-curves with one slope here but the method works similar for bilinear SN-curves in correspondence to Cosack (2011). Loads occurring during service life are tracked directly at the sensor

location and indirectly at other locations. This enables a comparison with (updated) design load calculations, from which



remaining useful lifetime can be derived. The algorithm also provides an estimate of the extrapolation uncertainty. This can be used for probabilistic assessment and potentially reduction of design safety factors.

Limitations are that an accurate simulation model is required. Updates are necessary whenever changes of structural properties occur (e.g. change of natural frequency due to scour). Reliability of strain sensors can be affected by temperature and other parameters. The method requires continues strain monitoring, so degradation of sensors over time might become problematic.

## 4 Conclusion and future work

This paper presents a method to extrapolate load measurements from one location to all hot spots of a monopile. Results are discussed for extrapolation from tower bottom to mudline. The algorithm was tested for other locations with comparable results (not shown here). We conclude that the correlation between DELs at different locations of the structure offers large potential for low-cost monitoring as only strain measurements at one level are needed. First tests show good accuracy of the suggested algorithm but further validation is necessary. The idea seems very promising and is highly recommended for further development. Future work should address:

- Validation with strain measurement data from two locations of a monopile.
- Sensitivity analysis of the extrapolation model to changes in structural properties.
- Detailed study on requirements of data resolution for calculation of DELs, number of simulations for extrapolation model, measurement duration.
- Integration of freely available SCADA data.

### Acknowledgement

We thank Anika Johansen and Daniel Kaufer (Ramboll) for input on monopile design. This project received funding from European Union's Horizon 2020 research and innovation program under the Marie Sklodowska-Curie grant agreement No 642108.

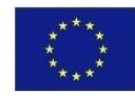

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
