# Peer review of "Structural monitoring for lifetime extension of offshore wind monopiles: Can strain measurements at one level tell us everything?"

_Wind Energy Science, 2017_

## Referee Comment (RC1) · Anonymous Referee #1 · 1 Jun 2017

*General comments:*

The paper presents a simple way of extrapolating damage (equivalent loads) to unmeasured hotspots on a monopile using only a single strain measurement. The method relies on aero-hydro-servo elastic simulations, an (updated) finite element model, and a statistical algorithm. The work is certainly considered worthy of publication. However, an effort should be made to present a more realistic estimate of the accuracy of the presented method by taking into account at least well-known error sources, e.g. noise on the measurements.

[Figure]

*Specific comments:*

- Potentially the largest errors come from situations in which the DELs calculated from the measured strains are inaccurate (since these form the basis for the extrapolation). This will be the case even for relatively low signal-to-noise ratios. It is suggested to investigate the effect of commonly encountered noise levels on the DEL calculation in order to give a more nuanced image of the accuracy that can be obtained with the proposed method.

- A review of existing literature on the subject is presented in the Introduction. It would be interesting to also see a comparison of the proposed methodology to the methods reviewed.

- It is presumed that the model updating mentioned on p2/line 23 is rather important for the accuracy of the estimates obtained with the proposed method. For this reason it might be good to give more detail regarding the proposed updating procedures.

- p8/line 1: "The algorithm also provides an estimate of the extrapolation uncertainty. This can be used for probabilistic assessment and potentially reduction of design safety factors." Care should be taken with statements like these since only uncertainties related to the ideal case (perfectly accurate structural model) are now considered.

*Technical corrections:*
p8/line 5: Please correct "continuous".

---

## Referee Comment (RC2) · Anonymous Referee #2 · 6 Jun 2017

**1 - General comments**

Lifetime extension (LTE) of offshore wind turbine support-structures is a topic with increased attention for wind farm operators and owners. More and more offshore wind farms are gradually approaching their design lifetime. Hence, the discussion about cost-efficient lifetime evaluation methods is of high interest and relevance to the industry. The manuscript approaches this field of interest by applying a combination of strain gauge measurement data at one location only, statistical methods and data simulations in a fatigue damage analysis. The method incorporates elements of traditional LTE analyses (based on pure strain gauge measurements) with model-based

simulations using statistical weather data. This combination ensures a close-to-reality analysis by using measurement data at the real asset, at the same time as it saves cost for extensive measurement campaigns by using simulation tools as a substitute. The manuscript is well written and recommended for publication subject to minor revisions.

The manuscript provides a good introduction to the approach itself and results obtained by evaluation of simulation data and predicted damage values. However, there is room for improvements by discussing several aspects in more detail:

- The analysis focuses on the prediction of fatigue damage values at mudline in the monopile. How critical is this location in the overall fatigue loading of the structure? Monopiles are commonly build by a pile structure with transition piece as connection to the tubular tower. How does the transition piece with grouting connection influence the ability to predict/transfer fatigue loading values from above to below the transition piece? Does other structural components influence the accuracy of the model in terms of prediction of values at other locations?

- Does the method include the possibility to evaluate fatigue of soil bearing capacity based on measured loads, or is this out of scope?

- What is the common practice for LTE analyses in the wind industry by today? How are such analyses performed for other offshore structures, e.g. oil and gas platforms? I'm aware of that loads on these structures are substantially different from what wind turbines experience. However, LTE is common practice in this area and experience from such analyses could potentially be transferred to the wind industry.

Several details of the performed analysis as well as technical solution are missing for an interested reader. Please address the following comments in a revised version:

- Include a matrix which shows load case combinations, or a reference to where the setup of the mentioned 'design basis and additional 1700 load cases' can be found.

- How does the technical solution of 'strain measurements at one level' look like? What

kind of strain gauges are required, how many and where are they placed? What kind of data do you extract from your simulation model at this level?

The manuscript shows fatigue damage prediction with very high accuracy (1-3%). To which extent will this accuracy be influenced by other potential errors as for example the fact that your simulation model must represent the real asset in detail? Degradation is mentioned in the discussion and needs to be accounted for in a measurement campaign. Noise of real world measurement data is another source of error and should be discussed and addressed in the approach.

2 - Specific comments

1 Introduction

- The BSH has a requirement that a CMS system has to be installed for at least 1/10 of the offshore wind turbines in a wind farm. Do you think the presented method could be a solution to fulfill such a requirement? Looked at it from a different angle, could an already installed support-structure CMS be used as input to your calculations and by this enable LTE calculations without additional sensor measurements?

- Traditional LTE analyses are for example based on strain gauge measurements for a certain period and statistical weather data. Do you know how many strain gauges need to be placed in traditional LTE measurement campaigns of monopiles? How large is the benefit for your solution, both in terms of reduced sensor costs as well as increased accuracy?

2 Methodology

- Please include the description of software programs used in the analysis (currently described in Section 3.1) in this section.

- SCADA-data are commonly recorded and are available for the lifetime of the asset. By retrofitting the proposed method in existing wind turbines, could historical SCADA-data in combination with experience from strain gauge measurements be used for an

evaluation of the fatigue damage experienced in the past?

3 Results and discussion

- It is not clear to the reader which number of neighbors you are ending up with, or which do you evaluate as sufficiently accurate. Figure 1 presents 4 neighbors on each side; Figure 10 presents 1 and 10 neighbors; and Table 1 and Figure 5 presents 1 and 15 neighbors. Please be more consistent on the data sets used and analyses performed to evaluate the sensitivity of the approach to different parameters.

- Have you evaluated to plot Table 1 as bar graph for better readability and direct comparison of the different approaches?

3 - Technical corrections

Page 1, line 19: 'better decisions', compared to what? Please rewrite.

Page 4, line 19: 'is predicted best', is resulting in highest accuracy? Please rewrite.

Page 5, figure 3: Please include grid lines in figure for better readability.

---

## Author Response (AR1)

**Response to Reviewer 1**

**Title of paper:** Structural monitoring for lifetime extension of offshore wind monopiles: Can strain measurements at one level tell us everything?

Dear Reviewer,

Thank you for the review of our submitted paper. We appreciate your constructive feedback and valuable comments on the topic. Please find below our suggestions, how we plan to modify the manuscript.

Please let us know if the suggested revisions fulfil your expectations.

Kind regards,

Lisa Ziegler - on behalf of the authors

03/07/2017

**Review comments**

The revision comments are organized as follows: The review comments are repeated in *italic*, responses are stated in normal black, and suggestion for revisions of the manuscript are shown in red.

**Specific comments**

*Comment 1: Potentially the largest errors come from situations in which the DELs calculated from the measured strains are inaccurate (since these form the basis for the extrapolation). This will be the case even for relatively low signal-to-noise ratios. It is suggested to investigate the effect of commonly encountered noise levels on the DEL calculation in order to give a more nuanced image of the accuracy that can be obtained with the proposed method.*

We plan to add a new test case in Chapter 3.1, which includes artificial measurement noise: For the new test case, artificial noise was imposed on the time series of bending moments at tower bottom extracted from the simulation model to represent potential measurement errors from strain sensors. The measurement noise was modelled as white Gaussian noise with zero mean and a signal-to-noise ratio of 40 dB. The procedure of rainflow counting and DEL calculation was performed equally to the previous test cases without artificial noise.

The extrapolation will then be performed as the following (Chapter 3.2 Extrapolation results): For the test case 'design with artificial measurement noise', the extrapolation model was calibrated with the computed T-DELs and M-DELs without noise. The noise affected 'measured' T-DELs were then used to predict corresponding M-DELs. Adding artificial noise on the simulated time series of bending moments at tower bottom increased the prediction error of lifetime M-DELs and damage by 1-2% in this case study.

We believe that further evaluation of sources of measurement noise and its magnitude is only meaningful with actual measurement data and out of scope of this brief communication paper. This should be investigated in future work.

*Comment 2: A review of existing literature on the subject is presented in the Introduction. It would be interesting to also see a comparison of the proposed methodology to the methods reviewed.*

We plan to add the following comparisons in Chapter 3.3 Discussion: Reference is made to Perisic and Tygesen (2014) for a comparison between existing approaches for structural health monitoring and our suggested approach. Perisic and Tygesen (2014) compare Kalman filter based methods and modal expansion for criteria including computational complexity, operation in real time, and structural model complexity. Kalman filter based methods have a low computational complexity, use reduced order FE models and can thus operate in real time. The complexity of structural models and computations for modal based algorithms is high resulting in an operation of near-real time (Perisic and Tygesen, 2014). Once the simulation data basis of the methodology presented here is set up, predictions can be performed

with almost no computational effort. This makes it possible to analyse large data sets in retrospect also. Algorithms based on artificial intelligence show similar computational performance. These algorithms, however, need sensors at every location for a training period. Perisic and Tygesen (2014) state that Kalman filter based methods and modal expansion perform similarly in terms of accuracy and sensitivity towards measurement noise. Future work with measurement data is needed to evaulate the sensitivity of the proposed methods to measurement noise.

Perišić N, & Tygesen UT. 2014. Cost-Effective Load Monitoring Methods for Fatigue Life Estimation of Offshore Platform. *In ASME 2014 33rd International Conference on Ocean, Offshore and Arctic Engineering.* American Society of Mechanical Engineers.

*Comment 3: It is presumed that the model updating mentioned on p2/line 23 is rather important for the accuracy of the estimates obtained with the proposed method. For this reason it might be good to give more detail regarding the proposed updating procedures.*

We also believe that the FE model updating is very important for good estimations with the method. We inserted a new Chapter 2.3 to give further information on this:

The process of FE model updating should verify that the global dynamic behaviour of the structure is captured correctly in the simulation model. Typical model updating techniques try to match natural frequencies, mode shapes, and damping. Devriendt et al (2014) use data from distributed accelerometers for operational modal analysis of on offshore wind turbine. Maes et al. (2016) show that the first and second fore-aft and side-side natural frequencies of a monopile are identifiable from strain gauge measurements at the tower in operating conditions of the wind turbine by transforming strain time series into power spectral densities. Modern turbines are often equipped with accelerometers in the nacelle whose measurements can be beneficial for the model updating procedure. After identification of the relevant modal properties, a sensitivity analysis should reveal which parameters in the original design model are uncertain and influential on the mismatched modal properties. For the case of the monopile support structure, these parameters can be, for instance, soil properties, manufacturing tolerances, grouted connection (early designs of transition pieces) and secondary steel elements if omitted in the initial FE model. Several methods exist to update the finite element model through minimization of an objective function addressing the selected parameters as described in standard literature (e.g. Friswell and Mottershead, 1995). The updating procedure should be repeated in time to identify possible changes on natural frequencies of the structure. Such changes could occur, for instance, due to scour or soil stiffening over time. Future work with measurement data is necessary to address FE model updating based on strain measurements for a monopile and the sensitivity of the extrapolation algorithm to this.

Devriendt, C., Weijtjens, W., El-Kafafy, M., & De Sitter, G. (2014). Monitoring resonant frequencies and damping values of an offshore wind turbine in parked conditions. *IET Renewable Power Generation, 8*(4), 433-441.

Friswell, M.I. and Mottershead, J.E. (1995). *Finite Element Model Updating in Structural Dynamics.* Netherlands: Kluwer Academic Publishers.

*Comment 4: p8/line 1: "The algorithm also provides an estimate of the extrapolation uncertainty. This can be used for probabilistic assessment and potentially reduction of design safety factors." Care should be taken with statements like these since only uncertainties related to the ideal case (perfectly accurate structural model) are now considered.*

Thanks for this comment. We plan to delete the two sentences since they might be misleading.

**Technical corrections**

Thanks for the technical corrections. We will implement this.

**Response to Reviewer 2**

**Title of paper:** Structural monitoring for lifetime extension of offshore wind monopiles: Can strain measurements at one level tell us everything?

Dear Reviewer,

Thank you for the review of our submitted paper. We appreciate your constructive feedback and valuable comments on the topic. Please find below our suggestions, how we plan to modify the manuscript.

Please let us know if the suggested revisions fulfil your expectations.

Kind regards,

Lisa Ziegler - on behalf of the authors

03/07/2017

**Review comments**

The revision comments are organized as follows: The review comments are repeated in *italic*, responses are stated in normal black, and suggestion for revisions of the manuscript are shown in red.

**General comments**

*Comment 1: The analysis focuses on the prediction of fatigue damage values at mudline in the monopile. How critical is this location in the overall fatigue loading of the structure? Monopiles are commonly build by a pile structure with transition piece as connection to the tubular tower. How does the transition piece with grouting connection influence the ability to predict/transfer fatigue loading values from above to below the transition piece? Does other structural components influence the accuracy of the model in terms of prediction of values at other locations?*

For the monopile design of our case study, the welds near mudline have a high fatigue life. Thus, these welds will not limit the overall fatigue life of the structure. The accuracy of the extrapolation method improves the smaller the distance between measurement location and predicted location is. We presented the extrapolation to mudline as an example, as this has a challenging distance to the measurement location. The method works comparably for other locations along the structure.

We plan to add this explanation in Chapter 3.3 Discussion: The extrapolation method is exemplarily presented here from tower bottom to mudline. The algorithm was tested for other locations with comparable results (not shown here). The accuracy of the extrapolation method improves the smaller the distance between measurement and predicted location is.

We used a flanged connection of the transition piece to tower and monopile in this case study. Only the early designs of monopiles and transition pieces still had full grouted connections. In these cases, the grouted connection is typically modelled by distributed connecting elements. The process of FE model updating should verify that the global dynamic behaviour of the structure is captured correctly in the model. If omitted secondary steel elements are important for the global dynamics, added masses shall be included during the FE model update. Future work with measurement data is necessary to address FE model updating and the sensitivity of the extrapolation algorithm to this.

We plan to add an explanation in Chapter 3.1 Case study: Turbine and monopile were modelled in detail following industry state-of-art. The turbine tower is connected to the monopile with a flanged transition piece.

Additionally, we plan to give additional information on FE model updating in the newly inserted Chapter 2.3: The process of FE model updating should verify that the global dynamic behaviour of the structure is captured correctly in the simulation model. Typical model updating techniques try to match natural frequencies, mode shapes, and damping. Devriendt et al (2014) use data from distributed accelerometers for operational modal analysis of on offshore wind turbine. Maes et al. (2016) show that the first and second fore-aft and side-side natural frequencies of a monopile are identifiable from strain gauge measurements at the tower in operating conditions of the wind turbine by transforming strain time series into power spectral densities. Modern turbines are often equipped with accelerometers in the nacelle whose measurements can be beneficial for the model updating procedure. After identification of

the relevant modal properties, a sensitivity analysis should reveal which parameters in the original design model are uncertain and influential on the mismatched modal properties. For the case of the monopile support structure, these parameters can be, for instance, soil properties, manufacturing tolerances, grouted connection (early designs of transition pieces) and secondary steel elements if omitted in the initial FE model. Several methods exist to update the finite element model through minimization of an objective function addressing the selected parameters as described in standard literature (e.g. Friswell and Mottershead, 1995). The updating procedure should be repeated in time to identify possible changes on natural frequencies of the structure. Such changes could occur, for instance, due to scour or soil stiffening over time. Future work with measurement data is necessary to address FE model updating based on strain measurements for a monopile and the sensitivity of the extrapolation algorithm to this.

Devriendt, C., Weijtjens, W., El-Kafafy, M., & De Sitter, G. (2014). Monitoring resonant frequencies and damping values of an offshore wind turbine in parked conditions. *IET Renewable Power Generation, 8*(4), 433-441.

Friswell, M.I. and Mottershead, J.E. (1995). *Finite Element Model Updating in Structural Dynamics.* Netherlands: Kluwer Academic Publishers.

*Comment 2: Does the method include the possibility to evaluate fatigue of soil bearing capacity based on measured loads, or is this out of scope?*

We expect that it is possible to identify changes in soil conditions as soon as these affect the natural frequencies of the structure. This is part of FE model updating, which is left for future work.
We plan to mention this briefly in the newly inserted paragraph 2.3 FE model updating: FE model updating can be used to identify changes that possibly occur in soil conditions over time, such as soil stiffening, as soon as these changes have a measurable effect on the natural frequencies of the structure.

*Comment 3: What is the common practice for LTE analyses in the wind industry by today? How are such analyses performed for other offshore structures, e.g. oil and gas platforms? I'm aware of that loads on these structures are substantially different from what wind turbines experience. However, LTE is common practice in this area and experience from such analyses could potentially be transferred to the wind industry.*

We have currently a paper submitted to a journal, which reviews the state-of-art of lifetime extension practises for onshore wind turbines in four European countries. For further details, we refer the interested reader to this paper.

We plan to add some explanation in Chapter 1 Introduction: There is almost no experience with lifetime extension of offshore wind turbines yet. Vindeby, the first commercial offshore wind farm installed in 1991, was decommissioned recently after 25 years of operation. Other existing structures, e.g. bridges, offshore oil platforms, and lately onshore wind turbines, have dealt with lifetime extension for multiple years already. Lifetime extension assessments and decision making in the oil and gas industry is discussed by Ersdal & Hörnlund (2008). Jackets for oil platforms are redundant structures where even the

loss of some members is often within acceptable limits of probability of failure. Lifetime assessments focus on detection of fatigue cracks in combination with fracture mechanics analyses. For offshore wind monopiles, however, Ziegler and Muskulus (2016c) have shown that the probability of detecting decisive fatigue cracks for lifetime extension of monopiles is small as the crack growth is expected to progress fast in the circumferential welds of these structures once it reaches a certain size. The authors conclude that numerical fatigue reassessment and structural monitoring is needed for lifetime extension decisions of monopiles. The state-of-art of lifetime extension in the onshore wind industry is reviewed by Ziegler et al. (submitted). Typically, lifetime extension assessments have an analytical and/ or practical part. The analytical part is a numerical fatigue reassessment where structural loading is recalculated with updated design models and assumptions (mainly environmental and operational conditions) (Ziegler and Muskulus, 2016). The practical part is on-site inspections, which would be possible but expensive due to offshore risks (Ziegler and Muskulus, 2016b).

Ersdal G, & Hörnlund E. 2008. Assessment of offshore structures for life extension. *In ASME 2008 27th International Conference on Offshore Mechanics and Arctic Engineering*. American Society of Mechanical Engineers.

*Comment 4: Include a matrix which shows load case combinations, or a reference to where the setup of the mentioned 'design basis and additional 1700 load cases' can be found.*

We plan to implement the following Table in Chapter 3.1:

| $V_W$ [m/s] | $H_S$ [m] | $T_P$ [s] | TI [%] | TI reduced [%] | IEC load case |
|---|---|---|---|---|---|
| 2-4 | 0.5-1.0 | 5.0-6.0 | 15-20 | 5-6 | 1.2, 6.4 |
| 5-8 | 0.5-1.5 | 5.0-6.0 | 15-17 | 4-5 | 1.2, 6.4 |
| 9-12 | 1.0-2.0 | 6.0-7.0 | 12-15 | 3-5 | 1.2, 6.4 |
| 13-16 | 2.0-3.0 | 6.5-7.5 | 10-12 | 3-4 | 1.2, 6.4 |
| 17-20 | 2.5-4.0 | 7.5-8.5 | 10-11 | 3-4 | 1.2, 6.4 |
| 21-24 | 4.0-5.0 | 8.5-9.5 | 10-11 | 3-4 | 1.2, 6.4 |
| 25-28 | 5.5-6.5 | 10.0-11.0 | 10-11 | 3-4 | 6.4 |
| 29-32 | 7.0-8.0 | 11.5-12.5 | 10-11 | 3-4 | 6.4 |

The load case combinations are presented in groups. Each group contains between 100-300 simulations of 10-minute duration with different random realizations (seeds). All wind directions (0-360°) are simulated in bins of 30° with two set of yaw errors. In addition, various wind-wave misalignments between 0-90° are considered for each wind direction.

*Comment 5: How does the technical solution of 'strain measurements at one level' look like? What kind of strain gauges are required, how many and where are they placed? What kind of data do you extract from your simulation model at this level?.*

We plan to add this explanation in Chapter 3.3 Discussion:

A feasible solution would be to install electrical resistance strain gauges at the upper part of the transition piece. The use of four axial strain gauges placed in 90° intervals around the circumferential is recommended. The redundancy of this setup enables to compare measurements from opposing strain gauges (compression and tension) to check the level of noise on the data. The sampling rate should be in the range of 20 Hz. The strain data must be calibrated and compensated for temperature. The time series of strain measurements can then be converted into bending stress or bending moments.

We plan to add this explanation in Chapter 3.1 Case study:

Time series of the bending moment around a local axis at a single point of the circumferential of tower (near tower bottom) and monopile (near mudline) were extracted from the simulations. The point of the circumferential would be chosen identical to the location of the strain gauges in a practical application.

**Specific comments**

*Comment 6: The BSH has a requirement that a CMS system has to be installed for at least 1/10 of the offshore wind turbines in a wind farm. Do you think the presented method could be a solution to fulfill such a requirement? Looked at it from a different angle, could an already installed support-structure CMS be used as input to your calculations and by this enable LTE calculations without additional sensor measurements?*

The BSH had a requirement of installing a CMS for 1/10 out the foundations in the standard BSH-No. 7005 'Design of offshore wind turbines' from 2007. However, this quantitative statement was removed in the new version of this standard from 2015 ('Minimum requirements concerning the constructive design of offshore structures within the Exclusive Economic Zone (EEZ)'). This new standard only specifies that '[…] parameters, such as shifting, deformation, component stress and frequencies, shall be measured and recorded (monitoring) in the area of the foundation elements at representative offshore wind turbine sites.'
We developed our methodology to be readily applicable to the sensor set ups which are already available in many projects (strain gauges at one level). Alternatively, it is also possible to install the necessary strain gauges in existing wind farms, as no work at submerged parts of the structure is required.

We suggest to not insert the details about the BSH standard into the manuscript since it is too much detail for the scope of the paper. However, we plan to clarify in Chapter 3.3 Discussion and Chapter 4 Conclusion that the necessary sensors are often already existing in many wind farms.

*Comment 7: Traditional LTE analyses are for example based on strain gauge measurements for a certain period and statistical weather data. Do you know how many strain gauges need to be placed in traditional LTE measurement campaigns of monopiles? How large is the benefit for your solution, both in terms of reduced sensor costs as well as increased accuracy?*

From what we have seen so far, the offshore wind industry has not found a consensus on a suitable, cost-effective way to apply lifetime extension analyses yet. This is mainly due to missing experiences and relevance as not many offshore wind farms are older than 15 years up to now. The main benefit of our proposed solution is that it can work with the sensors that many wind farms already have installed (strain gauges at one level). A study of sensor costs and a comparison of prediction accuracy between different structural monitoring approaches is out of scope of this paper and left for future research.

We plan to add a comparison between our suggested methodology to existing solutions for structural health monitoring in Chapter 3.3 Discussion: Reference is made to Perisic and Tygesen (2014) for a comparison between existing approaches for structural health monitoring and our suggested approach. Perisic and Tygesen (2014) compare Kalman filter based methods and modal expansion for criteria including computational complexity, operation in real time, and structural model complexity. Kalman filter based methods have a low computational complexity, use reduced order FE models and can thus operate in real time. The complexity of structural models and computations for modal based algorithms is high resulting in an operation of near-real time (Perisic and Tygesen, 2014). Once the simulation data basis of the methodology presented here is set up, predictions can be performed with almost no computational effort. This makes it possible to analyse large data sets in retrospect also. Algorithms based on artificial intelligence show similar computational performance. These algorithms, however, need sensors at every location for a training period. Perisic and Tygesen (2014) state that Kalman filter based methods and modal expansion perform similarly in terms of accuracy and sensitivity towards measurement noise. Future work with measurement data is needed to evaluate the sensitivity of the proposed methods to measurement noise.

*Comment 8: Please include the description of software programs used in the analysis (currently described in Section 3.1) in this section.*

Thanks for the comment. We plan to move this to a new Chapter 2.4 Case study.

*Comment 9: SCADA-data are commonly recorded and are available for the lifetime of the asset. By retrofitting the proposed method in existing wind turbines, could historical SCADA data in combination with experience from strain gauge measurements be used for an evaluation of the fatigue damage experienced in the past?*

Yes, we see the potential here to link the proposed method to historical SCADA data also. This would require a second step of training a model to predict DELs from input of SCADA.

We plan to add in Chapter 3.3 Discussion:

Many wind farms already have strain gauges installed at one level of the support structure. Alternatively, a retrofit of the necessary strain gauges is possible in existing wind farms, as no work at submerged parts of the structure is required. In case of retrofit, there is the potential to link the suggested extrapolation methodology to historical SCADA data (if recorded) in order to estimate the fatigue damage experienced in the past. This requires an additional model to infer DELs from SCADA and (possibly recorded) environmental conditions. This can be, for instance, a neural network algorithm as suggested by Smolka et al (2014).

*Comment 10: It is not clear to the reader which number of neighbors you are ending up with, or which do you evaluate as sufficiently accurate. Figure 1 presents 4 neighbors on each side; Figure 10 presents 1 and 10 neighbors; and Table 1 and Figure 5 presents 1 and 15 neighbors. Please be more consistent on the data sets used and analyses performed to evaluate the sensitivity of the approach to different parameters.*

Thanks for the comment. We plan to change Figure 1 and 2 for using 15 neighbors also to be consistent with the results presented later.

*Comment 11: Have you evaluated to plot Table 1 as bar graph for better readability and direct comparison of the different approaches?*

Thanks for the comment. We plan to transform the Table into a bar graph in the revised manuscript.

**Technical corrections**

Thanks for the technical corrections! We will implement this.

**Structural monitoring for lifetime extension of offshore wind monopiles: Can strain measurements at one level tell us everything?**

Lisa Ziegler[1,2], Ursula Smolka[1], Nicolai Cosack[1], Michael Muskulus[2]

[1]Ramboll Wind, 20097 Hamburg, Germany

[2]Department of Civil and Environmental Engineering, Norwegian University of Science and Technology NTNU, 7491 Trondheim, Norway

*Correspondence to*: Lisa Ziegler (lisa.ziegler@ramboll.com)

**Abstract.** Operators need accurate knowledge on structural reserves to decide about lifetime extension of offshore wind turbines. Load monitoring enables us to directly compare design loads with real loading histories of the support structure in order to calculate its remaining useful lifetime. Monitoring of every hot spot is technically and financially not feasible. This paper presents a novel idea for load monitoring of monopiles. It requires strain measurements at only one level convenient for sensor installation, such as tower bottom. Measurements are converted into damage equivalent loads for 10-minute time intervals. Damage equivalent loads are extrapolated to other locations of the structure with a simulation model and statistical algorithm. For this, structural loads at all locations of the monopile are calculated with aero-hydro-elastic software and updated finite element models. Damage equivalent loads at unmeasured locations are predicted from the simulation results with a k-nearest neighbor regression algorithm. The extrapolation was tested with numerical simulations of an 8 MW offshore wind turbine. Results show that damage can be predicted with an error of 1-3 % if this is done conditional on mean wind speed, which is very promising. The load monitoring concept is simple, cheap and easy to implement. This makes it ideal for taking decisions on lifetime extension of monopiles.

**1 Introduction**

Load monitoring of foundations for offshore wind turbines enables to reconstruct load histories that these structures experienced. The load history can be compared against design loads to calculate remaining useful lifetimes, which is essential for decisions on lifetime extension. Direct monitoring of every hot spot at the structure is impossible due to cost and access restrictions. Structural responses must be extrapolated from a limited set of sensors.

81% of offshore wind turbine foundations were monopiles in 2016 (Ho and Mbistrova, 2017). Existing monitoring strategies for monopiles are based on physical models or artificial intelligence. Model-based time-domain algorithms require accelerometers and (partly) strain gauges at the structure. They try to reproduce the time history of dynamic response parameters, such as acceleration or strain, of the whole structure. This has been investigated for monopiles using Kalman filters (Maes et al., 2016; Fallais et al., 2016), joint input-state estimation (Maes et al., 2016), and modal expansion algorithms (Maes et al., 2016; Iliopoulos et al, 2016).

In many cases, the remaining useful lifetime can be assessed using accumulated cycles or equivalent loads. Detailed load time series are not required. This is exploited by artificial intelligence algorithms (e.g. neural networks) (Smolka and Cheng, 2013; Cosack, 2011). After being trained using measurement data from all hot spots, the algorithms deduce statistics of dynamic response parameters, such as equivalent loads, from standard signals (Smolka et al, 2014).

5    There is almost no experience with lifetime extension of offshore wind turbines yet. Vindeby, the first commercial offshore wind farm installed in 1991, was decommissioned recently after 25 years of operation. Other existing structures, e.g. bridges, offshore oil platforms, and lately onshore wind turbines, have dealt with lifetime extension for multiple years already. Lifetime extension assessments and decision making in the oil and gas industry is discussed by Ersdal and Hörnlund (2008). Jackets for oil platforms are redundant structures where even the loss of some members is often within acceptable limits of
10   probability of failure. Lifetime assessments focus on detection of fatigue cracks in combination with fracture mechanics analyses. For offshore wind monopiles, however, Ziegler and Muskulus (2016a) have shown that the probability of detecting decisive fatigue cracks for lifetime extension of monopiles is small as the crack growth is expected to progress fast in the circumferential welds of these structures once it reaches a certain size. The authors conclude that numerical fatigue reassessment and structural monitoring is needed for lifetime extension decisions of monopiles. The state-of-art of lifetime
15   extension in the onshore wind industry is reviewed by Ziegler et al. (submitted). Typically, lifetime extension assessments have an analytical and/or practical part. The analytical part is a numerical fatigue reassessment where structural loading is recalculated with updated design models and certain assumptions (mainly environmental and operational conditions) (Ziegler and Muskulus, 2016b). Drawbacks are that long-term measurements of some environmental conditions, such as turbulence intensity, are often not available or expensive to obtain. The practical part is on-site inspections, which would be
20   possible but expensive due to offshore risks (Ziegler and Muskulus, 2016a). 
[revised manuscript text omitted]
 must be updated before the extrapolation can be performed (cf. Section 2). The process of FE model updating should verify that the global dynamic behaviour of the structure is captured correctly in the simulation model. Typical model updating techniques try to match natural frequencies, mode shapes, and damping. Operational modal analysis has been applied by Devriendt et al (2014) to identify natural frequencies and damping of an offshore wind turbine using accelerometers distributed at tower and transition piece. Modern turbines are often equipped with accelerometers in the nacelle. Additional accelerometers at the tower or transition piece are not always present. Maes et al. (2016) showed that the first and second fore-aft and side-side natural frequencies of a monopile are identifiable from strain gauge measurements at the tower in operating conditions of the wind turbine by transforming strain time series into power spectral densities.

After identification of the relevant modal properties, a sensitivity analysis should reveal which parameters in the original design model are uncertain and influential on the mismatched modal properties. For the case of the monopile support structure, these parameters can be, for instance, soil properties, manufacturing tolerances, grouted connection (early designs of transition pieces) and secondary steel elements if omitted in the initial FE model. Several methods exist to update the finite element model through minimization of an objective function addressing the selected parameters as described in standard literature (e.g. Friswell and Mottershead, 1995). The updating procedure should be repeated in time to identify possible changes on natural frequencies of the structure. Such changes could occur, for instance, due to scour or soil stiffening over time. Future work with measurement data is necessary to address FE model updating based on strain measurements for a monopile and the sensitivity of the extrapolation algorithm to this.

**2.4 Simulation software**

The software used for load simulations were LACflex and ROSAP (in-house tools of Ramboll). LACflex is an aero-elastic software for time-domain analysis of wind turbines based on the solver FLEX 5 (Passon and Branner, 2014). ROSAP is a structural analysis program which Ramboll uses for design of offshore wind foundations. The detailed model of the monopile is reduced to a Craig-Bampton superelement including corresponding wave loads for accurate integration into LACflex. Response time series at tower bottom are imported into ROSAP to model hydrodynamic loading and structural response of the detailed finite element model of monopile and transition piece (Passon and Branner, 2014; Passon, 2015). Design simulations of 10 min duration were performed for the fatigue load cases power production (DLC 1.2) and idling (DLC 6.4) according to IEC 61400-3 (IEC, 2009). Time series of the bending moment around a local axis at a single point of the circumferential of tower (near tower bottom) and monopile (near mudline) were extracted from the simulations. The point of the circumferential would be chosen identical to the location of the strain gauges in a practical application.

**3 Results and discussion**

**3.1 Case study**

Results are presented for a case study of a monopile in 30-40m water depth supporting an 8 MW wind turbine. Turbine and monopile were modelled in detail following industry state-of-art. The turbine tower is connected to the monopile with a flanged transition piece.

The extrapolation model is tested in five cases:

1. design basis,
2. extended simulations,
3. design basis with wind speed binning,
4. extended simulations with wind speed binning, and
5. design basis with artificial measurement noise.

The design basis includes 1700 load cases of normal operation and idling with wind speeds from 2 m/s to 32 m/s and corresponding sea states and turbulence intensity. Wind-wave directionality is considered in 30° bins including misalignment. The extended simulations include the design basis and additional 1700 load cases with reduced turbulence intensity. Table 1 presents the different load case combinations in groups. Each group contains between 100-300 simulations of 10-minute duration with different random realizations (seeds). All wind directions (0-360°) are simulated in bins of 30° with two sets of yaw errors. In addition, various wind-wave misalignments between 0-90° are considered for each wind direction.

**Table 1.** Load case combinations presented in groups consisting of mean wind speed $V_W$, significant wave height $H_S$, wave peak period $T_P$, turbulence intensity $TI$, and IEC load case. Each group contains 100-300 simulations of 10-minute duration.

| $V_W$ [m/s] | $H_S$ [m] | $T_P$ [s] | $TI$ [%] | $TI$ reduced [%] | IEC load case |
|---|---|---|---|---|---|
| 2-4 | 0.5-1.0 | 5.0-6.0 | 15-20 | 5-6 | 1.2, 6.4 |
| 5-8 | 0.5-1.5 | 5.0-6.0 | 15-17 | 4-5 | 1.2, 6.4 |
| 9-12 | 1.0-2.0 | 6.0-7.0 | 12-15 | 3-5 | 1.2, 6.4 |
| 13-16 | 2.0-3.0 | 6.5-7.5 | 10-12 | 3-4 | 1.2, 6.4 |
| 17-20 | 2.5-4.0 | 7.5-8.5 | 10-11 | 3-4 | 1.2, 6.4 |
| 21-24 | 4.0-5.0 | 8.5-9.5 | 10-11 | 3-4 | 1.2, 6.4 |
| 25-28 | 5.5-6.5 | 10.0-11.0 | 10-11 | 3-4 | 6.4 |
| 29-32 | 7.0-8.0 | 11.5-12.5 | 10-11 | 3-4 | 6.4 |

For the last test case, artificial noise was imposed on the time series of bending moments at tower bottom extracted from the simulation model to represent potential measurement errors from strain sensors. The measurement noise was modelled as white Gaussian noise with zero mean and a signal-to-noise ratio of 40 dB. The procedure of rainflow counting and DEL calculation was performed equally to the previous test cases without artificial noise.

M-DELs for the design basis (black) and lower turbulence intensity (red) are plotted in Figure 4 (left). The extended simulations follow the same pattern as the original set. In Figure 4 (right) M-DELs are colored according to their mean wind speed. The lower envelope observed for M-DELs is driven by wind speeds below rated power (12 m/s). The high end of the scatter occurs predominantly for load cases above cut-out wind speed (24 m/s).

[Figure]

**Figure 4.** M-DELs as function of ascending T-DELs. **Left:** M-DELs for design basis (black) and lower turbulence intensity (red). **Right:** Design basis M-DELs are colored according to the input mean wind speed.

**3.2 Extrapolation results**

Figure 5 and 6 present results of the extrapolation model in the five test cases. Results for the test cases 'design basis' $D$, 'design basis with wind speed binning' $D_{wind}$, 'extended simulations' $E$, and 'extended simulations with wind speed binning' $E_{wind}$ were obtained with leave-one-out cross-validation. For the test case 'design with artificial measurement noise' $D_{noise}$, the extrapolation model was calibrated with the computed T-DELs and M-DELs without noise. The noise affected 'measured' T-DELs were then used to predict corresponding M-DELs.

Figure 5 (left) shows a histogram of 1700 DEL ratios for the case 'design basis', where M-DELs are extrapolated as mean of 1 and 15 neighbors, respectively. It resembles the shape of a normal distribution for one neighbor, while the skewness increases for more neighbors, resulting in some overprediction of the average. Figure 5 (right) presents the variance of the M-DEL fractions 'predicted value / calculated value' for each test case, using 15 neighbors. Variance of the ratios is below 10% in all cases.

The percentage errors $e$ between predicted and calculated values for the target parameters, lifetime M-DELs and damage, are shown in Figure 6. Lifetime M-DELs and damage are extrapolated with errors smaller than 3% and 9% in the simulation environment. The extrapolation error is larger for damage than for lifetime DELs due to exponentiation with the material parameter $m$. Extrapolating $DEL^m$ instead of $DEL$ did not yield better results in this study (not shown). Weighting with the probability of occurrence and wind speed binning improves damage extrapolation, leading to errors smaller than 1% and 3%, respectively. Adding artificial noise on the simulated time series of bending moments at tower bottom increased the prediction error of lifetime M-DELs and damage by 1-2% in this case study.

[Figure]

[Figure]

**Figure 5. Left:** Histogram of DEL ratios for the case 'design basis' with 1 and 15 neighbors. **Right:** Variance of M-DEL ratios for all five test cases using 15 neighbors. The test cases are design basis $D$, design with artificial measurement noise $D_{noise}$, design basis with wind speed binning $D_{wind}$, extended simulations $E$, and extended simulations with wind speed binning $E_{wind}$. The neighbors are taken as mean (black) or weighted mean (grey).

[Figure]

[Figure]

**Figure 6.** Results for M-DELs with 15 neighbors in five test cases: design basis *D*, design with artificial measurement noise $D_{noise}$, design basis with wind speed binning $D_{wind}$, extended simulations *E*, and extended simulations with wind speed binning $E_{wind}$. The neighbors are taken as mean (black) or weighted mean (grey). **Left:** Percentage errors between predicted and calculated values of lifetime M-DELs. **Right:** Percentage errors between predicted and calculated values of lifetime damage.

**3.3 Discussion**

The novel idea for load monitoring is simple and easy to implement. No additional sensors, apart from strain measurements at one level, are needed. As an example, a technical solution would be to install electrical resistance strain gauges at the upper part of the transition piece. The use of four axial strain gauges placed in 90° intervals around the circumferential is recommended. The redundancy of this setup enables the comparison of measurements from opposing strain gauges (compression and tension) to check the level of noise on the data. The sampling rate should be in the range of 20 Hz. The strain data must be calibrated and compensated for temperature. The time series of strain measurements can then be converted into bending stress or bending moments.

Many wind farms already have strain gauges installed at one level of the support structure. Alternatively, a retrofit of the necessary strain gauges is possible in existing wind farms, as no work at submerged parts of the structure is required. In case of retrofit, there is the potential to link the suggested extrapolation methodology to historical SCADA data (if recorded) in order to estimate the fatigue damage experienced in the past. This requires an additional model to infer DELs from SCADA and environmental conditions. This can be, for instance, a neural network algorithm as suggested by Smolka et al (2014).

Reference is made to Perisic and Tygesen (2014) for a comparison between existing approaches for structural health monitoring and our suggested method. Perisic and Tygesen (2014) compare Kalman filter based methods and modal expansion for criteria including computational complexity, operation in real time, and structural model complexity. Kalman filter based methods have a low computational complexity, use reduced order FE models and can thus operate in real time. The complexity of structural models and computations for modal based algorithms is high resulting in an operation of near-real time (Perisic and Tygesen, 2014). Once the simulation data basis of the methodology presented here is set up,

predictions can be performed with almost no computational effort. This makes it possible to analyse large data sets in retrospect also. Algorithms based on artificial intelligence show similar computational performance. These algorithms, however, need sensors at every location for a training period. Perisic and Tygesen (2014) state that Kalman filter based methods and modal expansion perform similarly in terms of accuracy and sensitivity towards measurement noise. Future work with measurement data is needed to evaluate the sensitivity of the proposed methods to measurement noise.

Data transfer and storage of the presented method is efficient as the algorithm works with 10-minute values after conversion to DELs. The small costs of this solution make it feasible for application at every turbine in a wind farm. This removes the uncertainty of interpolation between turbines. DELs are calculated for SN-curves with one slope here but the method works similarly for bilinear SN-curves in correspondence to Cosack (2011). Loads occurring during service life are tracked directly at the sensor location and indirectly at other locations. This enables a comparison with (updated) design load calculations, from which remaining useful lifetime can be derived.

The extrapolation method is exemplarily presented here from tower bottom to mudline. The algorithm was tested for other locations with comparable results (not shown here). The accuracy of the extrapolation method improves the smaller the distance between measurement and predicted location is.

Limitations are that an accurate simulation model is required. Updates are necessary whenever changes of structural properties occur (e.g. change of natural frequency due to scour). Reliability of strain sensors can be affected by temperature and other parameters. The method requires continuous strain monitoring, so degradation of sensors over time might become problematic.

**4 Conclusion and future work**

This paper presents a method to extrapolate load measurements from one location to all hot spots of a monopile. Results are discussed for extrapolation from tower bottom to mudline as an example. We conclude that the correlation between DELs at different locations of the structure offers large potential for low-cost monitoring as only strain measurements at one level are needed. Many offshore wind farms have the necessary sensors and data already available making the developed method convenient for direct application. First tests show good accuracy of the suggested algorithm but further validation is necessary. The idea seems very promising and is highly recommended for further development. Future work should address:

- Validation with strain measurement data from two locations of a monopile.
- Sensitivity analysis of the extrapolation model to changes in structural properties.
- Detailed study on requirements of data resolution for calculation of DELs, number of simulations for extrapolation model, measurement duration.
- Integration of freely available SCADA data.

**Acknowledgement**

We thank Anika Johansen and Daniel Kaufer (Ramboll) for input on monopile design. This project received funding from European Union's Horizon 2020 research and innovation program under the Marie Sklodowska-Curie grant agreement No 642108.